# Biochemical Responses and Leaf Gas Exchange of Fig (*Ficus carica* L.) to Water Stress, Short-Term Elevated CO$_2$ Levels and Brassinolide Application

**Zulias Mardinata** [1,*] , **Tengku Edy Sabli** [2] **and Saripah Ulpah** [1]

1 Department of Agronomy, School of Graduate Studies, Islamic University of Riau, Marpoyan, Pekanbaru 28284, Indonesia; ulpahsaripah@agr.uir.ac.id
2 Department of Agrotechnology, Faculty of Agriculture, Islamic University of Riau, Marpoyan, Pekanbaru 28284, Indonesia; edysabli@agr.uir.ac.id
* Correspondence: zulias@agr.uir.ac.id; Tel.: +62-81371119313

**Abstract:** The identification of the key components in the response to drought stress is fundamental to upgrading drought tolerance of plants. In this study, biochemical responses and leaf gas exchange characteristics of fig (*Ficus carica* L.) to water stress, short-term elevated CO$_2$ levels and brassinolide application were evaluated. The 'Improved Brown Turkey' cultivar of fig was propagated from mature two- to three-year-old plants using cuttings, and transferred into a substrate containing 3:2:1 mixed soil (top soil: organic matters: sand). The experiment was arranged as a nested design with eight replications. To assess changes in leaf gas exchange and biochemical responses, these plants were subjected to two levels of water stress (well-watered and drought-stressed) and grown under ambient CO$_2$ and 800 ppm CO$_2$. Water deficits led to effects on photosynthetic rate, stomatal conductance, transpiration rate, vapour pressure deficit, water use efficiency (WUE), intercellular CO$_2$, and intrinsic WUE, though often with effects only at ambient or elevated CO$_2$. Some changes in content of chlorophyll, proline, starch, protein, malondialdehyde, soluble sugars, and activities of peroxidase and catalase were also noted but were dependent on CO$_2$ level. Overall, fewer differences between well-watered and drought-stressed plants were evident at elevated CO$_2$ than at ambient CO$_2$. Under drought stress, elevated CO$_2$ may have boosted physiological and metabolic activities through improved protein synthesis enabling maintenance of tissue water potential and activities of antioxidant enzymes, which reduced lipid peroxidation.

**Keywords:** biochemical responses; elevated CO$_2$; brassinolide; fig; water stress





## 1. Introduction

Water stress and excessively high temperature are two of the most common yield-limiting factors for crops in the world. The recognition of the key components to water stress is fundamental for upgrading plant drought tolerance [1–3]. To maintain their growth, plants have developed multiple tolerance tactics at the molecular, physio-biochemical, and morphological levels to react and adjust in water deficit conditions [2]. Plant drought resistance has four main strategies: drought avoidance, drought escape, drought tolerance, and drought recovery. The primary mechanisms include decreased water loss, improved water absorption, osmotic adjustment, osmoprotection, and increased antioxidant defenses [2,4].

Gases do not pass through the leaf cuticle but rather flow into and out of leaves via stomatal pores in the cuticle and epidermis, which are abundant on the lower surface of a leaf in most species. The stomata normally open during the day when the rate of photosynthesis is the highest. Physiological changes in the guard cells surrounding the stomatal pores account for their opening and closing [5]. Stomatal control is the first and most important step in response to drought, as stomatal conductance reduces the rate of water loss and slows the rate of water stress development, limiting its severity [6]. Stomatal

closure allows plants to restrict transpiration but it also limits $CO_2$ intake, which causes a decrement in photosynthetic rate [2,7]. Closing of stomata happens when two guard cells encircling the stomatal opening lose turgor pressure and close the pore [6]. An abscisic acid (ABA) signal is the most important signal inducing stomatal closure. The mechanisms of stomatal closure can be divided into active and passive forms. In passive closure, closure is adjusted by tissue water potential. In active closure, increasing ABA levels in leaves activate stomatal closure via ion exchange, concurrently increasing hydraulic conductivity simplifying ABA transport and water uptake in the roots [8].

The main greenhouse gas that contributes to the current global warming is $CO_2$ and it is crucial for plant growth [9]. In 2100, the concentration of atmospheric $CO_2$ is projected to reach 730–1020 µmol/mol. This means it will be higher than the current concentration as a consequence of a further increase in the cumulative emission of $CO_2$ into the atmosphere [10]. A greater amount of $CO_2$ in the air may improve photosynthetic rate as well as enhance biomass and yield of C3 and C4 crops directly, commonly known as the 'CO$_2$ fertilization effect'. Makino and Tadahiko [11] stated that leaf area expansion and N allocation into the rice (*Oryza sativa*) leaf were the main responses to elevated $CO_2$.

Amthor [12] divided the effects of $CO_2$ partial pressure on plant respiration into two categories: (1) direct effects observed during short-term changes in the $CO_2$ environment, and (2) indirect effects observed during long-term growth in a particular $CO_2$ partial pressure. $CO_2$ enrichment resulted in a more rapid leaf senescence which compounded an effect on acclimation. Kazemi et al. [13] showed that increasing $CO_2$ concentration by 200 mmol/mol during a rice (*Oryza sativa*) production season improved aboveground biomass significantly. Yang et al. [14] reported that $CO_2$ enrichment (570 mol/mol) hastened phenological development marginally (1–2 d) but significantly increased grain yield about 30% in rice. Dahal et al. [15] concluded that the responses of plants grown in an elevated $CO_2$ environment depend on both temperature and variety. It generally improved photosynthetic performance, water-use-efficiency, and grain yield in cereals.

An increase of $CO_2$ in the air influences many plants, especially in terms of photosynthetic rate and their growth which constitute more than 90% of terrestrial species, especially C3 and C4 plants. In general, elevated $CO_2$ increases the photosynthetic rate, and biomass production. However, long-term elevated $CO_2$ may lower the initial photosynthesis stimulation and suppress photosynthesis. Because the effects of $CO_2$ enrichment on photosynthesis are very complicated, many researchers have distinguished between the short-term and long-term effects of $CO_2$ [16].

The $CO_2$ assimilation rate is not only a function of stomatal opening, but non-stomatal (metabolic) limitation can occur during long-lasting drought stress. In particular, dry conditions can cause an increase of the xanthophyll pool, and a reduction in leaf absorbance, chlorophyll content, expression of ATP synthase, expression of *cyt b6/f*, and expression of functional proteins [17].

$CO_2$ enrichment increases starch content more than that of soluble sugars [18]. Some reports have shown that an intense augmentation of starch grains might cause physical damage to chloroplasts [19]. However, there is another possibility that starch accretion blocks $CO_2$ flow into the chloroplast [20]. A morphological alteration of chloroplasts in accumulating starch might be a crucial factor in $CO_2$ conductance because the conductance depends on the chloroplast surface area contiguous to the plasma membrane [21].

Though a drought-sensitive crop, fig (*Ficus carica* L.) is considered one of the most important crops in the world [22]. It needs 2.5 to 3.8 cm of water each week during the life cycle and can grow throughout the year [23]. To increase its productivity under water insufficiency, improving its tolerance against drought stress is imperative. Exogenous application of plant growth regulators to reduce the effects of water stress on crop plants is a pragmatic approach among various agronomic and physiological practices [24]. Natural and synthetic plant growth regulators have been extensively utilised for the induction of drought tolerance in crops [25]. Among the various compounds exploited to alleviate drought stress, brassinolide (BL) has been recognised for its ability to regulate plant growth

and productivity under water deficit conditions [26]. BL is known to alleviate various biotic and abiotic stress effects [27]. It has a unique growth-promoting action when applied exogenously [28].

To full characterize drought stress effects, it is important assess chlorophyll fluorescence, water use efficiency (WUE), proline accumulation, lipid peroxidation, and peroxidase (POD) and catalase (CAT) activities of stressed plants. Water stress in plants reduces the plant cell water potential and turgor, elevating solute concentrations. As a result, cell enlargement decreases leading to growth inhibition and reproductive failure. Many scientists believe that the first reaction of most plants to severe drought is the closure of their stomata to prevent the water loss via transpiration. This can impact WUE and gradually decreased PSII electron transport (chlorophyll fluorescence), triggering proline and malondialdehyde (MDA) accumulation, the latter from membrane lipid oxidative damage, and increasing the activity of enzymatic antioxidants POD and CAT.

To the best of our knowledge, few studies have been undertaken to investigate the potential of elevated $CO_2$ and BL application for alleviating water stress conditions in fig. The present study was conducted to understand the effects of short-term elevated $CO_2$ and BL application during water stress on biochemical responses and leaf gas exchange of fig.

## 2. Materials and Methods

### 2.1. Plant Material and Greenhouse Conditions

Cuttings from two to three year old mature 'Improved Brown Turkey' (IBT) fig trees were collected and transferred into containers containing 8 kg of 3:2:1 mixed soil (top soil: organic matter: sand). The propagated cuttings were watered frequently and remained in the shaded area for 1–2 weeks while rooting until they were ready to be shifted to several greenhouses in an experimental field at the School of Graduate Studies, Islamic University of Riau, Indonesia, located at 0°26′46.8″ N 101°27′21.8″ E in Marpoyan, Pekanbaru, Indonesia, from August 2018 to January 2019. The greenhouses were polyhouse-types with double spans, oriented west and east, covered with transparent UV stabilized polyethylene film with a 200-micron thickness covering the polyhouse roof and surrounded by plastic. Each house had two entrance doors, supplemented with one large circular fan inside and two medium circular fans on the left and right side of the roof, which were operated automatically based on the air temperature inside the greenhouse. During the experiment, the daytime mean air temperature was 24–30 ± 1 °C, the night-time mean air temperature was 17–22 ± 1 °C, and the daily mean relative humidity was maintained above 60 ± 2%.

### 2.2. Experimental Design

The experiment was arranged as a nested design with eight replications. The trees were placed under one of two $CO_2$ levels (elevated 800 ppm $CO_2$ or ambient $CO_2$). The two $CO_2$ levels were considered as the main fixed effects and two levels of water stress (well-watered [100% FC] and drought-stressed [25% of field capacity (FC)]) were considered as the random effects. There were four plants per treatment combination ($CO_2$ level by stress level). Data were recorded monthly for four months.

### 2.3. Application of Treatments

2.3.1. Brassinolide

Zulkarnaini et al. [29] reported that the best concentration of brassinolide to promote growth and physiological changes of fig was 200 mL·$L^{-1}$. Therefore this concentration was used. One-month-old fig tree seedlings were sprayed monthly with 200 mL·$L^{-1}$ BL solution (100 mL tetrahydroxymethyl-B-homo-oxacholestane lactone + 26 mL Multi Purpose Cultivation [MPC, Agrostar, SGR, Malaysia] + 20 L water) applied directly onto leaves between 900–1100 h.

2.3.2. Elevated $CO_2$

For treatment at 800 ppm $CO_2$, 34 kg $CO_2$ gas cylinders were injected using a nozzle with a flow rate of 0.5–5 L per minute automatically four times a day at 800, 830, 900, and 930 h for 8 min. After that, the two side doors were opened for air circulation and all of the circular fans were switched on automatically. $CO_2$ concentration, temperature, and relative humidity (RH) inside the polyhouses were monitored daily using a digital device ($CO_2$/Temp./RH Data Logger, Model AMF-102, Amtast, Taichung City, Taiwan).

2.3.3. Field Capacity Determination

Field capacity (FC) is the ability of soil particles to hold as much water as possible against gravity. A FC determination was conducted to assess the volume in a container at saturation. Seven 250 g pots were filled with 100 g of the planting media described above. All of the pots were watered with 100 mL daily until they were saturated, and then were left for 72 h until the water stopped dripping. The soils were then weighed as the wet weight (WW). The soils was put into the oven at 100 °C. After 24 h, the soil samples were removed from the oven, cooled in a desiccator, and then weighed as the dry weight (DW). To get the average result, the experiment was replicated 5 times. After that, the field capacity (FC) of the soil was calculated using the following formula [30]:

$$FC = (WW - DW)/DW \times 100 \tag{1}$$

The results indicated that field capacity was 38.3% or 38 mL. Thus, FC for 8 kg of planting media required a watering volume of 3040 mL and severe drought stress (25% FC) required a watering volume of 760 mL. The average water flow/pot was determined as 996 mL/min. Therefore the well-watered treatment (W0), or FC 100%, needed 3040 mL water, so was irrigated for 3 min/day. The severe drought stress treatment (W1), or FC 25%, needed 760 mL water and was irrigated for 1 min/day.

*2.4. Measurements*

2.4.1. Leaf Gas Exchange

Determination of Photosynthetic Rate (A), Stomatal Conductance (gs), Transpiration Rate (E), Intercellular $CO_2$ (Ci), and Vapour Pressure Deficit (VPD)

Measurements were obtained using an open system infrared gas analyser LICOR 6400 Portable Photosynthesis System (LICOR–6400, LI–COR Inc., Lincoln, NE, USA). Before use, the instrument was warmed for 30 min and calibrated with in ZERO IRGA mode. Two steps were required in the calibration process: first, an initial zeroing process for the built-in flow meter; and second, a zeroing process for the infrared gas analyser. The measurements used for optimal conditions were set by Evans and Santiago [31] at 400 ppm $CO_2$ at 30 °C cuvette temperature, 60% relative humidity and an airflow rate set at 500 $cm^3$ $min^{-1}$, and a modified cuvette condition of 800 ppm $CO_2$. The measurements of gas exchange were carried out between 900 to 1100 h using fully expanded young leaves at the third and/or fourth node from the plant apex. Photosynthetic rate (A), stomatal conductance (gs), transpiration rate (E), intercellular $CO_2$ (Ci), and vapour pressure deficit (VPD) were recorded. Several precautions were taken to avoid errors during measurements. Leaf surfaces were cleaned and dried using tissue paper before enclosing in the leaf cuvette. The leaf samples were placed on the cuvette for about one minute for data collection. Photosynthetically active radiation (PAR) ranged between 400–700 nm. The red 630 nm and blue 470 nm were modulated from 0.25 to 20 kHz. The operation was automatic and the data were stored in the LI-6400 console and analysed by the "Photosyn Assistant" software (Version 3, Lincoln Inc., Lincoln, NE, USA).

Determination of Water-Use Efficiency (WUE) and Intrinsic Water-Use
Efficiency (Int-WUE)

WUE refers to the ratio of A to water lost by the plant through transpiration. Photosynthetic water-use efficiency (also called intrinsic or instantaneous water-use efficiency) was defined as the ratio A to the rate of transpiration [32]:

$$WUE = A/E \qquad (2)$$

$$int\text{-}WUE = A/g_s \qquad (3)$$

where A = photosynthetic rate ($\mu mol \cdot m^{-2} s^{-1}$), $g_s$ = stomatal conductance ($mmol \cdot m^{-2} s^{-1}$) and E = transpiration rate ($mol \cdot m^{-2} s^{-1}$).

Determination of Chlorophyll Fluorescence

Measurements of chlorophyll fluorescence were taken from fully expanded young leaves. The measurement was obtained from 900 to 1200 h. The leaves were darkened for 15 min by attaching light-exclusion clips to the central region of the leaf surface. Chlorophyll fluorescence was measured using a portable chlorophyll fluorescence meter (Model Handy-PEA, Hansatech Instruments Ltd., King's Lynn, UK). Measurements were recorded for up to 5 s [33]. The fluorescence responses were induced by light emitting diodes. Measurements of initial fluorescence ($F_o$), maximum fluorescence ($F_m$), and variable fluorescence ($F_v$) were obtained from this procedure. $F_v$ was derived as the difference between $F_m$ and $F_o$, and the $F_v/F_m$ ratioj was calculated. The mean values of three representative plants per treatment combination were used to represent the data

2.4.2. Biochemical Response
Determination of Total Chlorophyll Content (T-Chl)

Fig leaves with different visible greenness (pale yellow, light green, and dark green) were selected for analysis. All samples were carried back to the laboratory, and the total leaf chlorophyll content was analysed. Leaf discs measuring 3 mm in diameter were obtained from each leaf sample using a hole puncher. The leaf disks were immediately immersed in 20 mL of 80% acetone in an aluminum foil-covered glass bottle and kept in the dark for approximately 7 days until all of the green colour had bleached out. Finally, 3.5 mL of the acetone extract solution was used to measure chlorophyll absorbance at two wavelengths using a light spectrophotometer (UV-3101P, Labomed Inc., Los Angeles, CA, USA). Two wavelengths—664 nm and 647 nm—were used to assess the peak absorbances of chlorophyll a and chlorophyll b, respectively. The total amount of chlorophyll a, chlorophyll b, and total amount chlorophyll were then calculated according to the method of Harris et al. [34]:

$$\text{Chlorophyll a content } (mg/cm^2) = 13.19 \, (A_{664}) - 2.57 \, (A_{647}) \qquad (4)$$

$$\text{Chlorophyll b content } (mg/cm^2) = 22.1 \, (A_{647}) - 5.26 \, (A_{664}) \qquad (5)$$

$$\text{Total Chlorophyll content } \left(mg/cm^2\right) = \frac{3.5 \times (\text{Chl a} + \text{Chl b})}{4} \qquad (6)$$

where $A_{647}$ and $A_{664}$ represent absorbance of the solution at 647 and 664 nm, respectively, while 13.19, 2.57, 22.1, and 5.26 were the absorption coefficients, 3.5 was the total volume used in the analysis taken from the original solution (mL) and 4 was the total disc area ($cm^2$).

Determination of Proline Content

Proline content of fully expanded leaves was determined according to Pesci and Beffagna [35]. Acid-ninhydrin reagent was prepared by warming 1.25 g ninhydrin in 30 mL glacial acetic acid and 20 mL of 6 M phosphoric acid, with agitation, until dissolved. The

reagent was stored at 4 °C, and remained stable for 24 h. Leaf samples (0.5 g fresh weight) were homogenised in 10 mL of 3% (*v/v*) aqueous sulfosalicylic acid and the homogenate was filtered through Whatman #2 filter paper. Two mL of filtrate was reacted with 2 mL of acid-ninhydrin and 2 mL of glacial acetic acid in a test tube and incubated in boiling water (100 °C) for 1 h. The reaction was terminated in an ice bath. The reaction mixture was extracted with 4 mL toluene, and shaken vigorously with a test tube stirrer for 15–20 s. The toluene layer at the top (pink-red) was collected with a pipette. The absorbance of the toluene layer was read at 520 nm using proline as standard. The proline concentration was determined from a standard curve and calculated on a fresh weight basis as follows:

$$\mu \text{ moles} \frac{\text{proline}}{\text{g}} \text{FW} = \frac{\left[ \frac{(\mu\text{g proline/mL} \times 4 \text{ mL toluene})}{115.5 \ \mu\text{g/}\mu\text{mole}} \right]}{0.5 \text{ g sample/5}} \tag{7}$$

Determination of Starch

Starch content was determined using the method of Chow and Landhausser [36]. In this method, about 250 mg of oven-dried tissue (at 45 °C until a constant weight, i.e., three days), was homogenized in 80% (*v/v*) ethanol to remove the sugar. The sample was then centrifuged at 5000× *g* for 5 min and the residue was retained. After that, 5 mL of cold distilled water and 6.5 mL of 52% perchloric acid was added to the residue. This solution was then centrifuged as above until the supernatant separated and filtered with Whatman #5 filter paper. The processes were repeated until the supernatant volume was 100 mL. A 100 μL aliquot of the supernatant was added to 900 μL distilled water. Then, 4 mL of anthrone reagent (200 mg of anthrone dissolved in 100 mL of 95% (*v/v*) sulphuric acid) was added to the 1 mL aliquot. The mixed solution was placed in a water bath at 100 °C for 8 min, cooled to room temperature, and the sample was read at an absorbance of 630 nm to determine the starch content. Glucose was used as a standard and starch content was expressed as mg glucose equivalent/g dry sample. A mean value of four replicate plants was used to represent each treatment combination.

Determination of Malondialdehyde (MDA) and Soluble Sugar Content (SSC)

Leaf tissue was assessed for malondialdehyde (MDA) and soluble sugar content (SSC) using the methods of Zhang and Huang [37]. Briefly, a 100 mg sample of fresh leaf tissue was ground in a mortar and pestle with 10 mL of 10% trichloroacetic acid ($C_2HCl_3O_2$) and a small quantity of quartz. The homogenate was centrifuged at 4000× *g* for 10 min, then a 2 mL aliquot was removed and mixed with 2 mL of 0.6% (*v/v*) thiobarbituric acid (TBA) solution. The solution was incubated at 100 °C for 15 min, allowed to cool, and then centrifuged again at 4000× *g*. Absorbance values of the supernatant were recorded at 532, 600, and 450 nm, and TBA was used as a standard. The MDA and SSC were calculated as follows:

$$\text{MDA} \left( \frac{\mu\text{mol}}{\text{g}} \text{FW} \right) = \frac{(6.45 \ (A_{532} - A_{600}) - 0.56A_{450})}{1000} \times \frac{\text{Vol. extrct solution (mL)}}{\text{Fresh Weight (g)}} \tag{8}$$

$$\text{SSC} \left( \frac{\text{mmol}}{\text{g}} \text{FW} \right) = \frac{(11.74 \ A_{450}) \ \text{mmol}}{1000\text{mL}} \times \frac{\text{Vol. extrct solution (mL)}}{\text{Fresh Weight (g)}} \tag{9}$$

Determination of Protein

The leaf protein contents were determined using bovine serum albumin (BSA) as a standard, according to the method of Bradford [38]. Briefly, fresh leaf samples (0.5 g) were ground with liquid $N_2$ in a mortar. The powder was mixed with 10 mL of 50 mM or 7.1 mL sodium phosphate buffer (pH 7.0) containing 1 mM or 0.3 g EDTA-Na2 and 2% (*w/v*) polyvinylpyrolidone (PVP) and homogenized. The homogenate was centrifuged at 11,000× *g* for 15 min at 40 °C. One milliliter of Bradford solution was added to 100 μL crude extract and the absorbance was recorded at 595 nm for an estimate of total protein

content. The leaf protein concentration was determined monthly and calculated from a BSA standard curve.

Determination of Peroxidase and Catalase

Enzyme activity assays of POD and CAT were based on the methods of Alici and Arabaci [39]. Fig leaves were harvested in the light, frozen in liquid $N_2$ and stored at $-80\,^{\circ}$C until assayed. To determine enzyme activities, leaf samples (0.5 g fresh weight) were ground with liquid $N_2$ in a mortar and pestle. The homogenate powder was mixed with 10 mL of 50 mM or 7.1 mL sodium phosphate buffer (pH 7.0) containing 1 mM or 0.3 g of EDTA-Na$_2$ and 2% (*w/v*) polyvinylpyrrolidone (PVP). The extract was centrifuged at $16,000\times g$ for 4 min at $4\,^{\circ}$C, and the supernatant was used for the following enzyme assays.

Peroxidase (POD) activity was assayed by measuring the ability of the enzyme extract to increase absorption at 470 nm due to the oxidation of guaiacol. A reaction mixture consisting of 3 mL of 0.1 M sodium phosphate buffer (pH 7.0), 28 µL guaiacol, and 30 µL of 30% $H_2O_2$ was prepared and incubated at $32\,^{\circ}$C. Then, 100 µL of enzyme extract was added to the mixture and the increase in absorbance at 420 nm was measured for at least 2 min with readings at 30 s intervals to determine absorbance change at 0.01 as POD activity.

Catalase (CAT) activity was assayed by measuring the ability of the enzyme extract to decompose $H_2O_2$. The reaction mixture consisted of 2 mL of 0.1 M sodium phosphate buffer (pH 7.0), 1 mL of 0.08% (*v/v*) $H_2O_2$, and 0.2 mL enzyme extract. One unit of CAT activity measured at 405 nm was defined as 1 µmol $H_2O_2$ consumed per mg of tissue protein s$^{-1}$. The CAT activity was expressed as enzyme units per mg of protein.

*2.5. Statistical Anakysis*

Monthly means were calculated from the collected data and were analyzed using analysis of variance (ANOVA) by Statistical Analysis System (SAS 9.4, SAS Institute Inc., Cary, NC, USA) to determine significant differences. Differences between treatments means were compared by using Fisher's Least Significant Difference (LSD) at $p \leq 0.05$ levels. All figures and tables are presented as the mean $\pm$ standard deviation. Data are presented from those months after treatment (MAT) when significant differences were first noted.

## 3. Results

*3.1. Leaf Gas Exchange*

3.1.1. Photosynthetic Rate, Stomatal Conductance, Transpiration Rate, Intercellular $CO_2$, and Vapour Pressure Deficit

Leaf gas exchange of *Ficus* sp. was affected by water stress within $CO_2$ levels of *Ficus* sp. (Figure 1). Drought stress led to a decline in photosynthesis (A) and intercellular $CO_2$ in ambient conditions, and stomatal conductance (gs) at high $CO_2$, and increased stomatal conductance and vapour pressure deficit (VPD) at ambient conditions compared to well-watered conditions.

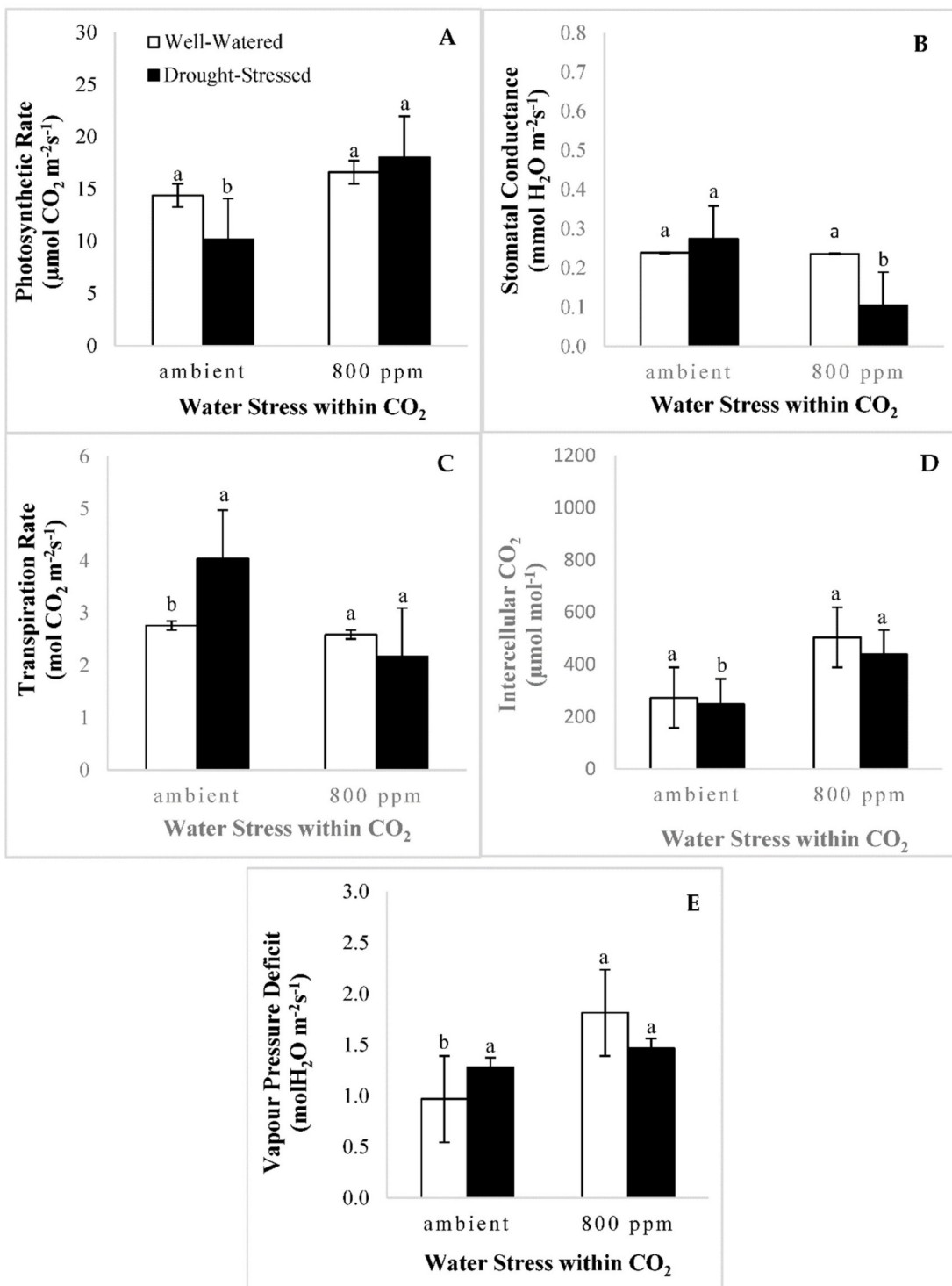

**Figure 1.** Effect of water stress within $CO_2$ levels on *Ficus* sp. with brassinolide application according to parameters: (**A**) at first month after treatment (MAT), (**B**) gs at second MAT, (**C**) E at second MAT, (**D**) Ci at second MAT, and (**E**) VPD at third MAT. Bars represent means ± SD. Bars with different letters were significantly different by Fisher's LSD at $p < 0.05$.

### 3.1.2. Water-Use-Efficiency, Intrinsic WUE, and Chlorophyll Fluorescence

WUE was reduced by high $CO_2$ and more by drought stress (Figure 2A). Intrinsic WUE was reduced by drought at ambient $CO_2$ only. The chlorophyll fluorescence ($F_v/F_m$) decreased with drought stress only at elevated $CO_2$.

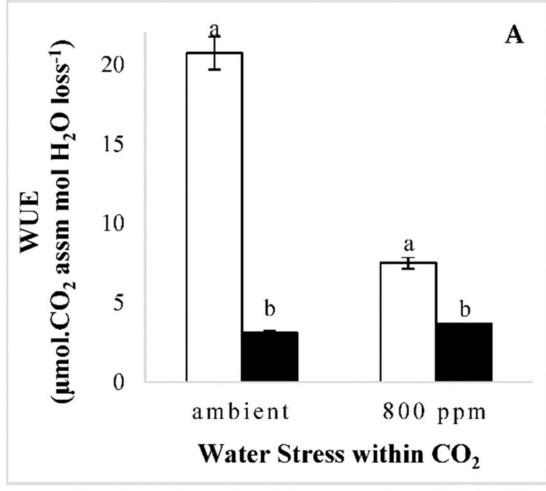

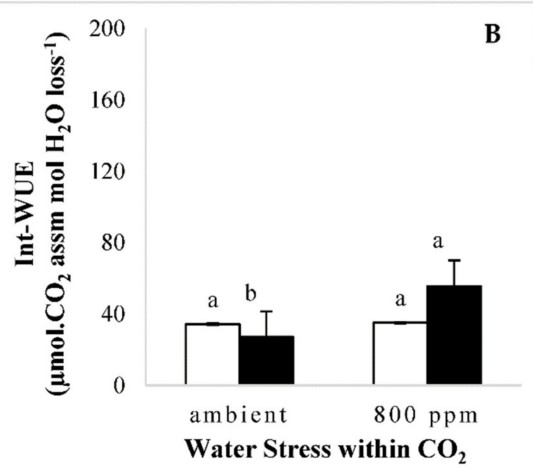

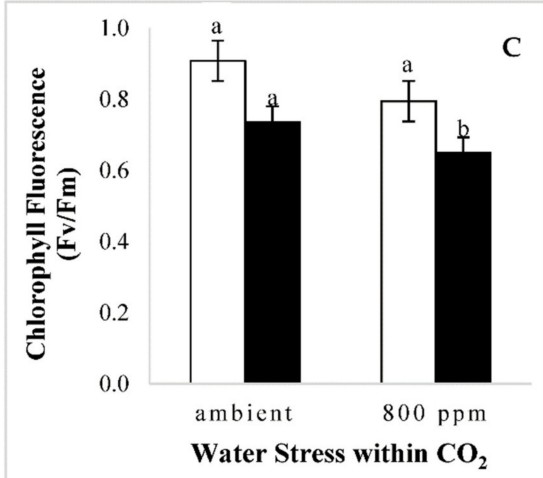

**Figure 2.** Effect of water stress within $CO_2$ levels with brassinolide application on *Ficus* sp. leaf gas exchange: (**A**) WUE treatment of water stress within $CO_2$ at third MAT, (**B**) int−WUE treatment of water stress within $CO_2$ at first month after treatment (MAT), and (**C**) $F_v/F_m$ treatment of water stress within $CO_2$ at third MAT. Empty bars are well−watered, and black bars are drought stressed. Bars represent means ± SD. Bars with different letters were significantly different by Fisher's LSD at $p < 0.05$.

### 3.2. Biochemical Responses

#### 3.2.1. Chlorophyll Content, Lipid Peroxidation, Osmolyte Accumulation, Protein, and Starch Content

In drought conditions, malondialdehyde (MDA) increased at both $CO_2$ levels, proline and soluble sugar content (SSC) increased at ambient $CO_2$ only, whereas starch and protein content decreased at ambient $CO_2$ but total chlorophyll content decreased at high $CO_2$ (Figure 3).

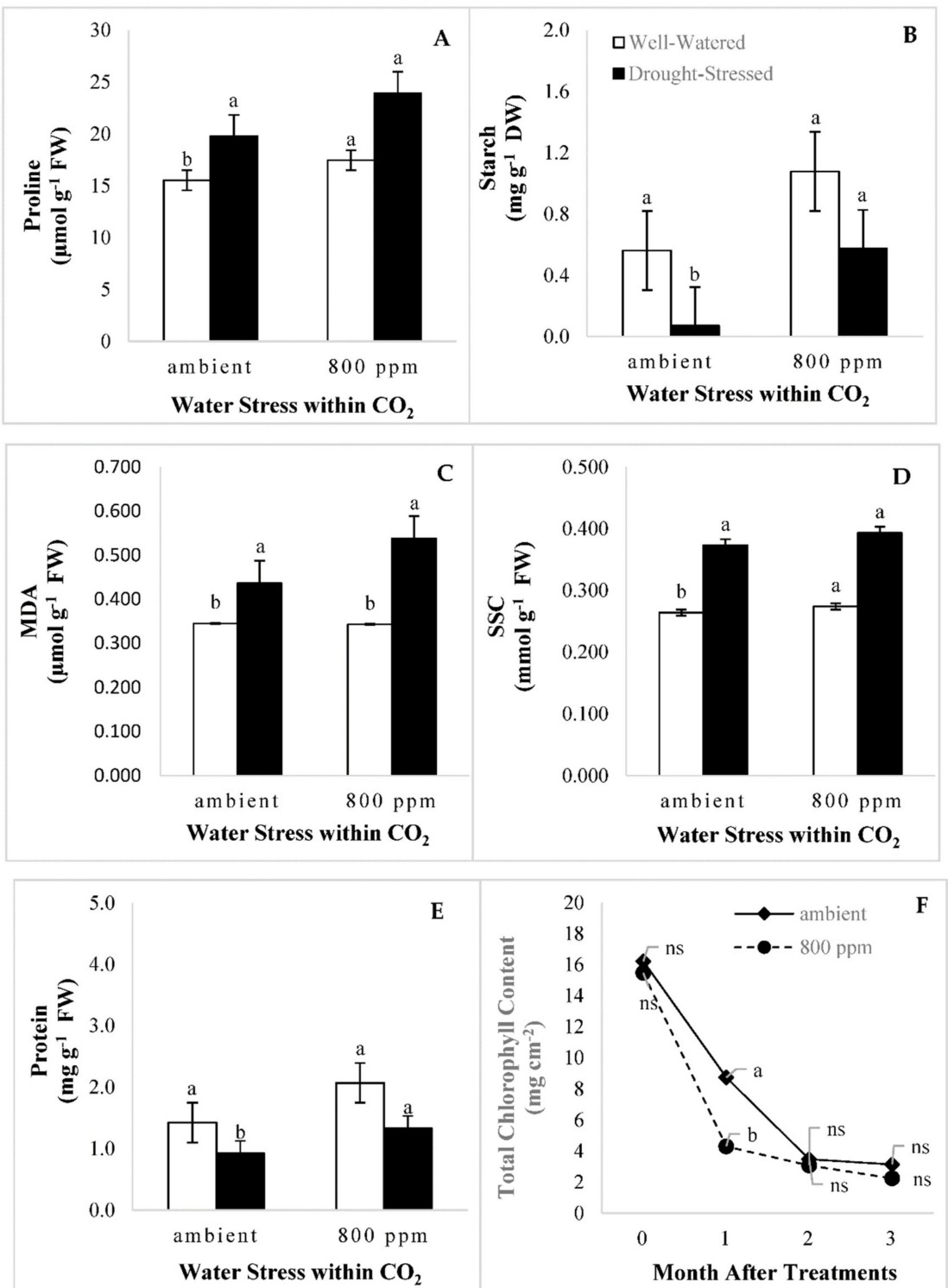

**Figure 3.** Effect of water stress within $CO_2$ levels with brassinolide application on *Ficus* sp. (**A**) proline content at second month after treatment (MAT), (**B**) starch at third MAT, (**C**) MDA at third MAT, (**D**) SSC at third MAT, and (**E**) protein content at third MAT, and (**F**) T−Chl as the main effect of water stress. Bars and curves (solid line is well−watered and dashed line is drought stressed) represent means ± SD. Bars with different letters were significantly different by Fisher's LSD at $p < 0.05$.

### 3.2.2. Activation of Antioxidant Defense Systems

The drought led to significant changes in antioxidant defenses in leaves of *Ficus* sp. plants. POD activity increased in *Ficus* sp. plants due to the imposition of drought with elevated $CO_2$ only (Figure 4). Enriched $CO_2$ conditions caused a further increase in antioxidant enzyme activities of the plants.

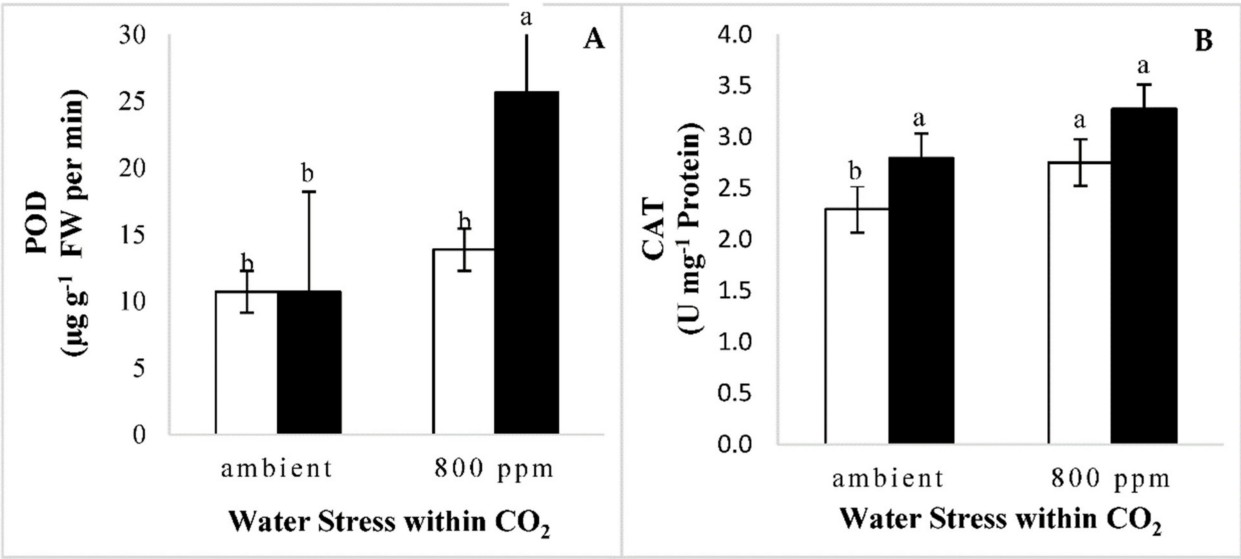

**Figure 4.** Effect of water stress within $CO_2$ levels and brassinolide application on *Ficus* sp. (**A**) POD at third month after treatment (MAT), and (**B**) CAT at third MAT. Bars represent means ± SD. Empty bars are well−watered, and black bars are drought stressed. Bars with different letters are significantly different by Fisher's LSD at $p < 0.05$.

### 3.3. Correlation Analysis

A correlation analysis was carried out to establish the relationship between the quantitative parameters. Table 1 shows significant positive correlations among parameters such as A with protein; gs with $F_v/F_m$; VPD with WUE; WUE with $F_v/F_m$; int-WUE with proline; $F_v/F_m$ with starch; proline with MDA; proline with POD; MDA with POD; etc. The meaning of the positive correlation between proline and MDA was both were significantly increased with stress.

A significant negative correlation was noted between gs with POD; E with WUE; Ci with VPD; VPD with POD; WUE with SSC; int-WUE with protein; $F_v/F_m$ with MDA; T-Chl with proline; proline with starch; MDA with protein; SSC with starch; protein with CAT; etc. The meaning of the negative correlation between E with WUE was likely due to both an increase in E as well as a decrease in WUE.

**Table 1.** Pearson correlation between all measured parameters in the experiment. * and ** indicate significant at $p < 0.05$ and 0.01, respectively.

| | A | gs | E | Ci | VPD | WUE | Int-WUE | Fv/Fm | T_Chl | Proline | MDA | SSC | Protein | POD | CAT |
|---|---|---|---|---|---|---|---|---|---|---|---|---|---|---|---|
| A | 1 | | | | | | | | | | | | | | |
| gs | 0.132 | 1 | | | | | | | | | | | | | |
| E | 0.235 | 0.206 | 1 | | | | | | | | | | | | |
| Ci | 0.443 ** | −0.011 | 0.192 | 1 | | | | | | | | | | | |
| VPD | −0.155 | 0.444 ** | −0.122 | −0.351 * | 1 | | | | | | | | | | |
| WUE | 0.188 | 0.114 | −0.675 ** | −0.067 | 0.335 * | 1 | | | | | | | | | |
| Int-WUE | 0.226 | −0.619 ** | 0.016 | 0.101 | −0.271 | −0.061 | 1 | | | | | | | | |
| Fv/Fm | 0.176 | 0.333 * | −0.130 | −0.090 | 0.539 ** | 0.400 ** | −0.032 | 1 | | | | | | | |
| T_Chl | −0.012 | 0.436 ** | −0.032 | −0.133 | 0.512 ** | 0.244 | −0.216 | 0.262 | 1 | | | | | | |
| Proline | 0.006 | −0.438 ** | 0.248 | −0.063 | −0.419 ** | −0.236 | 0.324 * | −0.074 | −0.303 * | 1 | | | | | |
| MDA | −0.241 | −0.656 ** | −0.157 | −0.044 | −0.377 ** | −0.187 | 0.257 | 0.018 | −0.428 ** | 0.462 ** | 1 | | | | |
| SSC | −0.161 | −0.583 ** | 0.076 | −0.003 | −0.413 ** | −0.357 * | 0.269 | 0.037 | −0.279 | 0.362 * | 0.642 ** | 1 | | | |
| Protein | 0.370 ** | 0.651 ** | 0.265 | 0.243 | 0.252 | 0.131 | −0.436 ** | −0.429 ** | 0.149 | −0.234 | −0.520 ** | −0.506 ** | 1 | | |
| POD | −0.064 | −0.210 | 0.118 | −0.016 | −0.308 * | −0.140 | 0.196 | −0.040 | −0.100 | 0.315 * | 0.436 ** | 0.425 ** | −0.132 | 1 | |
| CAT | −0.131 | −0.584 ** | −0.078 | 0.078 | −0.589 ** | −0.202 | 0.234 | −0.117 | −0.498 ** | 0.522 ** | 0.769 ** | 0.746 ** | −0.403 ** | 0.563 ** | 1 |
| Starch | 0.432 ** | 0.740 ** | 0.208 | 0.216 | 0.396 ** | 0.226 | −0.342 * | −0.475 ** | 0.224 | −0.304 * | −0.637 ** | −0.649 ** | 0.866 ** | −0.249 | −0.556 ** |

## 4. Discussion

### 4.1. Leaf Gas Exchange

The reaction and durability of plants to water scarcity are a coalition of intricate biological processes happening at the molecular, cellular, physio-biochemical, and whole plant levels [40]. Drought tolerant plants have developed some mechanisms to adjust to water deficits like organized alteration in growth, photosynthetic rates, osmotic adaptation and energy metabolism, senescence and cell death, and metabolic modifications [41].

Plant growth and photosynthetic performance can be significantly blocked by drought stress as a consequence of the disorder or restriction of normal physiological and metabolic processes. Reduced growth can be recognized as a growth adjustment tactic to water scarcity in plants [42]. In this research, some parameters of leaf gas exchange were significantly affected, assuming that drought stress caused the loss of water from plant cells and growth restriction.

Many previous researchers found similar results in other drought stresses species [41,43]. Restrictions in stomatal and non-stomatal conductances indicated that there was photosynthetic inhibition in some species [44,45]. Our results indicated that some gas exchange and water use traits were affected 1 month after treatment initiation, implying changes in photosynthetic performance under drought stress. However, high $CO_2$ levels often altered the responses compared to ambient $CO_2$ levels. In contrast some responses such as the malondialdehyde content increased due to drought stress at both $CO_2$ levels. This might be caused by inequality between electron requirement for photosynthesis and photochemical activity at PS II under drought stress, triggering the photo-damage of the photosynthetic systems. This is turn leads to the overabundance of ROS in green plant cells and intensifying oxidative injury to cells [46].

We found that chlorophyll fluorescence values ranging between 0.65 and 0.91. These $F_v/F_m$ values were comparable to previous research. Wang et al. [47] investigated $F_v/F_m$ values in *Ficus tikoua* leaves and showed that chlorophyll fluorescence values were 0.53–0.76; $F_v/F_m$ values on *Ficus carica* were 0.81–0.93 [48], whereas $F_v/F_m$ values on *Ficus benjamina* were 0.72–0.79 [49]. Moreover, values measured in healthy plants ranged from 0.7 to 0.85 [33]. $F_v/F_m$ measurements enable the identification of alterations in the common bioenergetics state of the photosynthetic tools. Furthermore, such measurements relate, directly or indirectly, to all steps of light-dependent photosynthetic reactions, including electron transport, water splitting, the pH gradient establishment across the thylakoid membrane, and ATP synthesis [50]. In another study, initial chlorophyll fluorescence $(F_o)$ rose significantly under drought stress, and the maximum photochemical efficiency $(F_v/F_m)$, as well as photochemical quenching coefficient (qP) decreased resulting in the decline of the quantum yield of PSII (ΦPSII) [51]. In addition, low electron transport through PSII and the loss of PSII activity $(F_v/F_m)$ resulted in a decline in photosynthesis.

It was evident that the mean VPD was higher in the greenhouse with elevated $CO_2$ than ambient $CO_2$ (Figure 1E). This was because the same air entered all of the greenhouses and they were not controlled for humidity. Warm air can hold more water vapour so the relative amount of vapour was low. The lower rate of transpiration may have been due to stomatal closure at a high VPD and high temperature (a direct effect of elevated $CO_2$) [52].

The down-regulation of photosynthesis by enhancement of $CO_2$ can be observed when the photosynthate content surpasses the carbohydrate used for growth. Nonetheless, many plant species exhibit little or no photosynthetic down-regulation although the plants were grown with long-term enhancement of $CO_2$ and have surplus carbohydrates. For example, radish (*Raphanus sativus*) [53] and potato (*Solanum tuberosum*) [54] did not show any photosynthetic down-regulation whilst they genetically belong to starch-accumulating species. Tubers or roots can play a role as a significant sink for carbohydrates. For radishes, the storage root biomass distinctly increased under enhanced $CO_2$, and therefore no excess carbohydrate accumulation was detected in the leaves [53]. Likewise in rice, the sheaths of a leaf can play a role as momentary carbohydrate sinks, and the number of absolute carbohydrates in the leaf blades may be markedly low [55]. Thus, the downregulation

of photosynthesis in rice might be small compared to bean (*Phaseolus* spp.) [54], cotton (*Gossypium hirsutum* L.) [56], or soybean (*Glycine max* L.) [57]. These plants gather a great deal of carbohydrates in the leaves, specifically as starch in chloroplasts. The difference with *Ficus* sp. may be because it is a C4 plant and does not have tubers or roots which can act as a large carbohydrate sink like potato and radish, therefore elevated $CO_2$ caused no downregulation of photosynthesis in *Ficus* sp. Additionally, the application of brassinolide may have had an impact on increased physiological and biochemical responses of *Ficus* sp. [29,58,59].

Various short-term responses to elevated $CO_2$ might correlate to differences in the whole plant sink-source relations depending on the developmental stages. Elevated $CO_2$ always caused a higher excitation of biomass production in young plants than in mature seedlings like soybean [57], cotton [60], tobacco (*Nicotiana tabacum*) [61], and rice [62].

Photosynthetic pigment (carotenoid) accumulation may occur during the drought response in *Ficus* sp. [63]. Carotenoids, also called tetraterpenoids, are yellow, orange, and red organic pigments that are produced by plants and algae, as well as several bacteria and fungi. *Ficus* sp. contained all major carotenoids, albeit at low concentrations [64]. Our data again found that the *Ficus* sp. could more adequately stimulate the activities of antioxidant enzymes, relating to the adjustment to long-term dryness.

The key photosynthetic pigment is chlorophyll. The results showed that the T-Chl increased when enriched with $CO_2$ and decreased with drought stress at some times likely due to the T-Chl reflecting leaf photosynthesis ability and plant health status [65]. Greater $CO_2$ availability was associated with T-Chl, and therefore an increase in photosynthetic rate.

### 4.2. Biochemical Responses

Plants gather osmolytes under drought stress, like soluble sugars, proteins, and amino acids (proline) (Figure 5) to maintain cell turgor [2]. The level of osmolyte acquisition was different under water deficits. Our results showed that soluble sugars and proline content all increased under water deficits at ambient $CO_2$ levels. It implied that osmolyte acquisition has a fundamental task in *Ficus* sp. acclimation to dryness through maintaining cell turgor and cell membrane function. This then took care of the macromolecules from injury caused by water deficits [4,66]. Turkan et al. [67] and Sultan et al. [68] reported a greater proline content acquisition during drought of *Phaseolus acutifolius* and wheat (*Triticum aestivum* L.) variety (BG-25) with PEG-simulated water deficits. Furthermore, *Ficus* sp. showed greater SSC at elevated $CO_2$ than at ambient levels, indicating that stressed *Ficus* sp. may osmoregulated by the acquisition of osmolytes to enhance water holding with long-term water scarcity.

The short-term $CO_2$ enhancement effects on photosynthesis have been varied. Commonly, extended exposure to $CO_2$ enhancement decreases photosynthesis in the beginning in many plant species and may inhibit photosynthetic rate. These reactions have been assigned to secondary reactions relevant to redundant carbohydrate acquisition pathways. Carbohydrate acquisition in *Ficus* sp. leaves may cause the inhibition of photosynthetic gene expression and surplus starch appears to inhibit $CO_2$ dissipation. Thus, plants will not show photosynthetic inhibition if they don't have sink organs for carbohydrate acquisition [11]. Various photosynthetic downregulation events amongst many species could be tightly correlated to a divergence in their quantity of starch accumulation in leaves [69].

A negative effect of elevated $CO_2$ may be reduced tissue nutrient concentration, particularly nitrogen, which results in lower grain protein content [70]. High temperatures increase the rates of transpiration and, hence, crop demand for water [71]. Under severe drought stress, plants with the ability to adjust osmotically can maintain turgor when leaf water potential is reduced, to sustain leaf gas exchange, cellular membrane and protein function, as well as chloroplast volume and function [72].

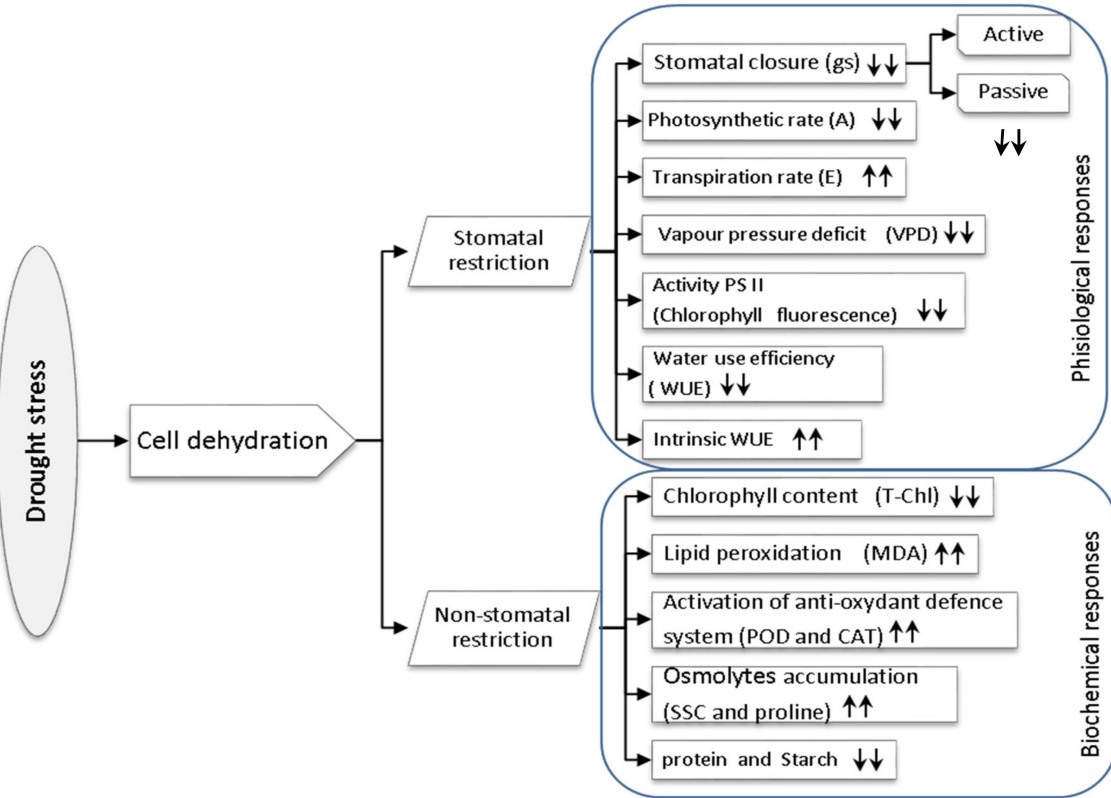

**Figure 5.** Drought tolerance mechanism flowchart on physiological and biochemical responses basis in plants. Source: developed [73,74].

For a self-defense system, most species have a protective enzyme-catalysed cleanup system [74]. In case the plants endure a water deficit, all of the defensive systems need to be stimulated to limit active oxygen damage. MDA is a damage product of the decomposition of polyunsaturated fatty acids, and the MDA content shows the lipid peroxidation level caused by oxidative injury [75]. Rising MDA content under water scarcity as in the present work implies that dryness might increase reactive oxygen species (ROS) and cause lipid peroxidation [76]. Furthermore, increasing MDA even at elevated $CO_2$ conditions is considerably more prominent than with $CO_2$-treatment alone in *Ficus* sp.

Drought triggered the fig antioxidant enzyme activity, including POD and CAT, implying that antioxidant enzymes might be responsible for *Ficus* sp. protection during water scarcity. Consistent with the present results, an increase in the POD and CAT activities have been reported in various soybean cultivars [41]. Nevertheless, long-term dryness had no effect on the activities of POD and CAT. In contrast, Xu et al. [77] found that under expanded water scarcity conditions, the POD and CAT activities in Kentucky bluegrass (*Poa pratensis*) significantly declined. Reduced antioxidant enzyme activity implies that the capacity to scavenge ROS is reduced, which might cause ROS-mediated oxidative injury, including membrane lipid peroxidation.

## 5. Conclusions

*Ficus carica* grown under drought stress after brassinolide application often had different responses under elevated $CO_2$ than ambient conditions. Fewer differences between well-watered and drought-stressed plants were evident at elevated $CO_2$ than at ambient $CO_2$. The levels of malondialdehyde, SSC, and proline may play a role in drought resistance in fig.

**Author Contributions:** Z.M. and S.U. designed the experiments and analyzed the data; Z.M., T.E.S., and S.U. performed the experiments and analyzed the data; Z.M. and T.E.S. wrote the manuscript. All authors have read and agreed to the published version of the manuscript.

**Funding:** This research received no external funding.

**Institutional Review Board Statement:** Not acceptable.

**Informed Consent Statement:** Not acceptable.

**Data Availability Statement:** Not acceptable.

**Conflicts of Interest:** The authors declare no conflict of interest.

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
