# Peer review of "Biochemical Responses and Leaf Gas Exchange of Fig (Ficus carica L.) to Water Stress, Short-Term Elevated CO2 Levels and Brassinolide Application"

_horticulturae, doi:10.3390/horticulturae7040073_

Round 1
Reviewer 1 Report
Experimental design
In this part, the number of replications should be clarified, as the abstract says (line 16-17) "The experiment was arranged as a Nested design with four replications." but in the Material and Method 2.2 part (line 126) that "The experiment was arranged as a Nested design with eight replications."
Methods/ Measurements
The part considering the description of measurements should be changed and reworded to a small extent to avoid similarity with previous publication. Otherwise, the methods are adequately described.
Results
In this part, the textual evaluation of the results is adequate, however, I miss the number of measurements (n = .......) and (mean+/- S.D.) from the description of the figures. Also, the notation of the standard deviation values is missing from the top of the columns of the result figures in all the cases. These should be corrected.
Author Response
Comments 1:
The manuscript titled “ Biochemical Responses and Leaf Gas Exchange of Fig (Ficus carica L.) under Water Stress, Short-Term Elevated CO2 and Optimized Brassinolide Concentration” deals with the investigation of water stress effect and elevated CO2 response on Ficus carica. The manuscript is well written thought there is one point to be properly discussed. At L 537 538 Why under drought stress in control plants there is an increase of transpiration and a decrease of photosynthesis? (Fig 1) panel b and c are consistent between each other but contrast with panel a and d! this is not consistent with what is discussed at L533
Response 1 :
There are some reasons:
First, Ficus carica plant is a C3 plant. C3 plants are more adaptif at high level CO2 concentration. It mean C3 plants can benefit from the elevated levels of CO2 (Marry and Donald,1997; Driscol et al., 2006). At ambient CO2 level (control plants), photosynthesis in C3 plants is inhibited by a process known as photorespiration. When this process occurs, about half of the photosynthetic yield is lost and returned to the air (Arcelia and Maribel, 1999). During photorespiration, oxygenation and carboxylation reactions take place at the same active site of Rubisco, decreasing the efficiency of photosynthesis in C3 plants (Coleman and Sage, 2001). C3 plants have higher photorespiration than C4 plants.
Second, the plant materials in this research are two to three years matured fig. It means that figs are during a growing season. In growing season at control plants, a fig leaf will transpire many times more water than its own weight especially at drought stress plants. At same situations, the effect will different at elevated CO2. In addition, all of fig samples have been giving optimized brassinolide concentrations that will increase the growth and physiological responses of fig.
Comments 2:
Furthermore in the introduction the physiological background of stomata regulation under drought is not included. In particular the role of ABA and the interaction with passive mechanism of stomata closure. You can take example of these in the work done on grapevine, for a review see Marusig and Tombesi 2020, https://doi.org/10.3390/ijms21228648..
Response 2 :
The introduction part has been revised and highlighted. The role of ABA and the interaction with passive mechanism of stomata closure have been added with reference from Marusig and Tombesi 2020.
Comments 3:
L179 LI6400 is an open system, please amend the text
Response 3 :
The word at Line 179 has been changed to “an open system”.
Comments 4:
Fig 1 the measuring unit of Stomatal conductance is wrong, it should be mol H2O m-2 s-1
Response 4 :
The unit of Stomatal conductance has been changed to “mol H2O m-2.s-1”
Reviewer 2 Report
The manuscript titled “ Biochemical Responses and Leaf Gas Exchange of Fig (Ficus carica L.) under Water Stress, Short-Term Elevated CO2 and Optimized Brassinolide Concentration” deals with the investigation of water stress effect and elevated CO2 response on Ficus carica. The manuscript is well written thought there is one point to be properly discussed. At L 537 538 Why under drought stress in control plants there is an increase of transpiration and a decrease of photosynthesis? (Fig 1) panel b and c are consistent between each other but contrast with panel a and d! this is not consistent with what is discussed at L533
Furthermore in the introduction the physiological background of stomata regulation under drought is not included. In particular the role of ABA and the interaction with passive mechanism of stomata closure. You can take example of these in the work done on grapevine, for a review see Marusig and Tombesi 2020, https://doi.org/10.3390/ijms21228648..
Other minor issues
L179 LI6400 is an open system, please amend the text
Fig 1 the measuring unit of Stomatal conductance is wrong, it should be mol H2O m-2 s-1
Author Response
Reviewer 1 :
Comments 1:
Experimental design
In this part, the number of replications should be clarified, as the abstract says (line 16-17) "The experiment was arranged as a Nested design with four replications." but in the Material and Method 2.2 part (line 126) that "The experiment was arranged as a Nested design with eight replications."
Response 1 :
The abstract part has been revised and highlighted. The number of replications has been changed to “eight replications”.
Comments 2:
Methods/ Measurements
The part considering the description of measurements should be changed and reworded to a small extent to avoid similarity with previous publication. Otherwise, the methods are adequately described.
Response 2 :
No action is taken with some reasons :
- Measurements of leaf gas exchange cannot be spiltted with small extent because the data of photosynthesis rate, stomatal conductance, transpiration rate, intercellular CO2, and vapour pressure deficit have been taken in once action.
- The other measurements, authors have been grouped based on their close correlation therefore it will simplified the results
Comments 3:
Results
In this part, the textual evaluation of the results is adequate, however, I miss the number of measurements (n = .......) and (mean+/- S.D.) from the description of the figures. Also, the notation of the standard deviation values is missing from the top of the columns of the result figures in all the cases. These should be corrected.
Response 3 :
The Results part have been revised and highlighted. The number of measurements (n=…) has been added to description of figures and also the notations in the top of column have been added with SD values.
Reviewer 2:
Comments 1:
The manuscript titled “ Biochemical Responses and Leaf Gas Exchange of Fig (Ficus carica L.) under Water Stress, Short-Term Elevated CO2 and Optimized Brassinolide Concentration” deals with the investigation of water stress effect and elevated CO2 response on Ficus carica. The manuscript is well written thought there is one point to be properly discussed. At L 537 538 Why under drought stress in control plants there is an increase of transpiration and a decrease of photosynthesis? (Fig 1) panel b and c are consistent between each other but contrast with panel a and d! this is not consistent with what is discussed at L533
Response 1 :
There are some reasons:
First, Ficus carica plant is a C3 plant. C3 plants are more adaptif at high level CO2 concentration. It mean C3 plants can benefit from the elevated levels of CO2 (Marry and Donald,1997; Driscol et al., 2006). At ambient CO2 level (control plants), photosynthesis in C3 plants is inhibited by a process known as photorespiration. When this process occurs, about half of the photosynthetic yield is lost and returned to the air (Arcelia and Maribel, 1999). During photorespiration, oxygenation and carboxylation reactions take place at the same active site of Rubisco, decreasing the efficiency of photosynthesis in C3 plants (Coleman and Sage, 2001). C3 plants have higher photorespiration than C4 plants.
Second, the plant materials in this research are two to three years matured fig. It means that figs are during a growing season. In growing season at control plants, a fig leaf will transpire many times more water than its own weight especially at drought stress plants. At same situations, the effect will different at elevated CO2. In addition, all of fig samples have been giving optimized brassinolide concentrations that will increase the growth and physiological responses of fig.
Comments 2:
Furthermore in the introduction the physiological background of stomata regulation under drought is not included. In particular the role of ABA and the interaction with passive mechanism of stomata closure. You can take example of these in the work done on grapevine, for a review see Marusig and Tombesi 2020, https://doi.org/10.3390/ijms21228648..
Response 2 :
The introduction part has been revised and highlighted. The role of ABA and the interaction with passive mechanism of stomata closure have been added with reference from Marusig and Tombesi 2020.
Comments 3:
L179 LI6400 is an open system, please amend the text
Response 3 :
The word at Line 179 has been changed to “an open system”.
Comments 4:
Fig 1 the measuring unit of Stomatal conductance is wrong, it should be mol H2O m-2 s-1
Response 4 :
The unit of Stomatal conductance has been changed to “mol H2O m-2.s-1”